# The Role of Cell Wall Polysaccharides Disassembly and Enzyme Activity Changes in the Softening Process of Hami Melon (*Cucumis melo* L.)

**DOI:** 10.3390/foods11060841

**Published:** 2022-03-15

**Authors:** Weida Zhang, Minrui Guo, Wanting Yang, Yuxing Liu, Yue Wang, Guogang Chen

**Affiliations:** College of Food Science and Technology, Shihezi University, Shihezi 832000, China; zwd9411@163.com (W.Z.); 18139280260@163.com (W.Y.); liuyuxing233@163.com (Y.L.); 17590396006@163.com (Y.W.); cgg611@163.com (G.C.);

**Keywords:** Hami melon, cell wall composition, fruit softening, cell-wall-modifying enzymes, cell wall polysaccharides

## Abstract

To investigate the physiological and molecular properties relating to cell wall carbohydrate metabolism in fruit, the ultrastructure and polysaccharides compositions of the cell wall, as well as the fruit quality and activities of enzymes relating to fruit softening, were studied for three Hami melon varieties (‘Xizhoumi 17’, ‘Jinhuami 25’, and ‘Chougua’) representing three different storability levels. The results showed that ‘Chougua’ maintained a higher firmness on day 18, with the lowest decay incidence (0%). ‘Chougua’ showed a better storage quality and intact cell wall structure. The molecular weight and monosaccharide composition of cell wall polysaccharides for Hami melons underwent great changes during storage, and the degradation of pectin polysaccharides was obvious, involving the depolymerization of macromolecular polymers accompanied by the production of new macromolecular polymers and composition changes in pectin monosaccharides (glucose, galactose, and arabinose) during the softening process of the Hami melons. Polygalacturonase, pectin methylesterase, xyloglucan endo-transglycosylase/hydrolase, α-arabinofuranosidase, β-galactosidase, and cellulase were associated with fruit softening at different stages of storage. There were similar softening mechanisms in the three Hami melons. This study will provide reference for further study on the fruit softening mechanisms of Hami melons.

## 1. Introduction

Fruit softening during postharvest storage is closely related to changes in cell wall structure and composition, and it tends to have great negative influences on fruit appearance, texture, flavor, and aroma [1]. Studies have shown that the cell wall disassembly and middle lamella dissolution during the fruit softening process can attenuate cell–cell adhesion. Further, multiple complex physiological and biochemical changes occur during the process [1,2,3,4]. Wei et al. (2015) and Chea et al. (2019) found that the hemicellulose, pectin, and cellulose in cell wall have obvious changes during fruit ripening, and the synergistic action of cell-wall-modifying enzymes, mainly including polygalacturonase (PG), β-galactosidase (β-GAL), α-arabinofuranosidase (α-ARF), pectin methylesterase (PME), xyloglucan endo-transglycosylase/hydrolase (XTH), and cellulase (Cx), are also involved in the process.

Hami melon (*Cucumis melo* L.) is a climacteric fruit which ripens and undergoes senescence rapidly after harvest, with has serious decay loss. It has been favoured by customers for years because of its sweet flesh and abundant nutrition [5]. Harvested Hami melons ripen and soften quickly if they are stored at room temperature [5], and their sensory quality also decreases, which greatly reduce the melon’s shelf life. Undoubtedly, the poor storability of Hami melon has caused great economic losses. At present, most studies focus on the preservation of Hami melon by edible coating [1], enhancing storage quality, and extending the shelf life of fresh-cut Hami melon [6]. However, few reports explore the internal relationships among cell wall composition, cell-wall-modifying enzymes, and fruit softening of Hami melon.

‘Xizhoumi 17’ (*C. melo* var. *inodorus Jacq*., XZM 17), ‘Jinhuami 25’ (*C. melo* var. *inodorus Jacq.*, JHM 25), and ‘Chougua’ (*C. melo* var. *inodorus*., CG) are three landraces of Hami melon with different storability in Xinjiang, China. Among them, the ‘XZM 17’ and ‘JHM 25’ fruit tended to be soft when they were stored at room temperature, while the quality of ‘CG’ was obviously better than that of the former two, with only slight shrivelling and softening at the end of storage [1]. To explore the reasons for the differences in the softening of the three landraces, the alterations in ultrastructure of cell walls, cell wall composition, firmness, and softening-associated enzyme activity were measured to make clear the relationship between cell wall solubilization and fruit softening and the changes in the physiological properties among Hami melon landraces. This study will provide a theoretical basis for controlling fruit softening and make contributions to the shelf-life extension and quality decline inhibition of postharvest Hami melons.

## 2. Materials and Methods

### 2.1. Experimental Materials

‘XZM 17’, ‘JHM 25’, and ‘CG’ (Appendix A), three widely cultivated landraces of Hami melon in Xinjiang, were collected from local orchards in Hami, Xinjiang, China, and delivered to the lab in Shihezi University within eight hours. Fruits of similar size, shape, and color, with no mechanical damage or disease, were selected and stored at 20 ± 1 °C at 25% RH (relative humidity) for 18 d. The weights of the selected ‘XZM 17’, ‘JHM 25’, and ‘CG’ fruits were 1555~1580 g, 2470~2490 g, and 1080~1100 g, respectively. The length and diameter of the Hami melon fruit were measured using calipers (Mitutoyo, Kawasaki, Japan). The lengths were 18~20 cm, 24~27 cm, and 17~19 cm, respectively, and the diameters were 12~14 cm, 12~13 cm, and 11~12 cm, respectively. Fifty-five fruits of one landrace were used for analysis. Sampling and measurements were performed every three days. Fruit pulp was quickly frozen in liquid nitrogen and then stored at −80 °C for later use.

### 2.2. Determination of Soluble Solid Content and Ascorbic Acid Content

Soluble solid content (SSC) was determined following the method of Kumar et al. (2017). The SSC in Hami melon juice was measured with the GMK-701R digital refractometer (G-Won Hitech, Seoul, Korea), and the results were expressed as percentage of total soluble solids (%). The ascorbic acid (AsA) content was measured with 2, 6-dichlorophenol indophenol [7,8] and the result was expressed as g kg^−1^ on a fresh weight basis.

### 2.3. Examization of Fruit Firmness, Weight Loss, Decay Incidence, and Respiration Rate

The firmness of the Hami melons was determined with a GY-4 sclerometer (Zhejiang Tuopuyunnong Scientific Instruments Co., Ltd., Hangzhou, China). The Hami melons were peeled and the maximum force (N) was recorded. Three fruits were measured at every measuring date, and each Hami melon’s firmness was determined three times at the equator position. Then, the average value was calculated.

The weight loss (*WL*) of the Hami melons was determined with an electronic balance and calculated according to the method of Kumar et al. (2017), as follows:(1)$$WL\;\left(\%\right)=\frac{OW-FW}{OW}\times 100\;$$
where *OW* is the original weight (g) of the Hami melon and *FW* is the final weight (g) of the Hami melon on the sampling date.

Decay incidence was determined according to the method of Raynaldo et al. (2021). Twenty fruits of each landrace were randomly selected to measure decay incidence. The respiration rate was detected as CO_2_ production. A 9 L glass jar with Hemi melon (approximately 2500 g) was enclosed for 30 min prior to CO_2_ sampling, the gas was detected with an infrared gas analyzer (IR400, Yokogawa Co., Ltd., Shanghai, China), and the results of the respirations rate were expressed as ng kg^−1^ s^−1^ [9]. All measurements were conducted three times.

### 2.4. Ultrastructure of the Cell Wall

The pulp of Hami melon fruit (1 cm^3^) was collected and the samples were pretreated according to the method of Fan et al. (2019). Pulp of the equatorial part of Hami melons was fixed in glutaraldehyde solution (2.5%) overnight, and then the sample was treated as follows: ① the sample was rinsed with phosphoric acid buffer solution (0.1 M, pH 7.0) three times for 15 min each time; ② the sample was fixed (1%) with osmium acid solution for 1–2 h; ③ the samples were rinsed with phosphoric acid buffer (0.1 M, pH 7.0) three times for 15 min each time; ④ the samples were dehydrated with 50%, 70%, 80%, 90%, 95%, and 100% ethanol for 15 min each, and then treated with acetone for 20 min; ⑤ the samples were treated with a mixture of 1:1 and 3:1 embedding agent and acetone successively for 1 h and 3 h; ⑥ the samples were treated with pure embedding agent overnight; and ⑦ after the osmotic treatment of the sample was buried, and the embedded sample was heated at 70 °C overnight.

The samples were sliced on a LEICAEM UC7 ultra-thin slicer to obtain sections of 70–90 nm, which were passed through lead citrate solution and uranium dioxane acetate 50% ethanol saturated solution and stained for 15 min each. Finally, the samples were observed under a transmission scanning electron microscope (H-7650, accelerating voltage 100 kV, Hitachi Co., Ltd., Beijing, China) (TSEM).

### 2.5. Extraction of Cell Wall Material

Cell wall material (CWM) was extracted in the form of alcohol-insoluble residue (AIR), according to the method of Rose et al. (1998). Flesh of the Hami melons was pulped with a homogenizer (SPCH-10, Homogenizing Systems Ltd., Harlow, UK). Then, anhydrous ethanol in the amount of 0.15 L was added to the flesh (50 g) in a beaker (covered with plastic wrap), followed by a boiling water bath for 20 min to destroy the cell wall and inactivate enzymes. After filtration, the residue was soaked in mixed chloroform–methyl alcohol solution (1:1, *v*/*v*) in the amount of 0.10 L for 20 min. The mixture was repeatedly washed using acetone in the amount of 0.25 L until the filtrate was colorless. To remove starch in the residue, the residue was stirred in a 90% dimethylsulfoxide (DMSO) solution in the amount of 0.1 L overnight. Subsequently, the residue was washed twice by an 80% ethanol solution (0.2 L), dried to a constant weight at 60 °C, and weighed as AIR. The results were expressed as g kg^−1^ in a fresh-weight basis.

### 2.6. Separation of Cell Wall Polysaccharides

AIR (500 mg) was extracted with 30 mL of distilled water at room temperature for 4 h by constant oscillation. The extract was filtered, centrifuged, and lyophilized (FD-1A-50, Shanghai Hefan Instruments Co., Ltd., Shanghai, China) as a water-soluble fraction (WSF). Then, the residue was added to 30 mL of a 50 mM cyclohexylenedimaine tetraacetic acid (CDTA)-acetate buffer (pH = 6) solution at 25 °C with constant oscillation for 4 h. After filtration, the residue was extracted once again by buffer solution. Both of the extract solutions were mixed and dialyzed (Mw cut off 3500 Da) at room temperature for 48 h. Finally, the freeze-dried solution was obtained as a chelate-soluble fraction (CSF). In the same way, the residue was successively extracted with 30 mL of 50 mM Na_2_CO_3_ (containing 20 mM NaBH_4_) to obtain a sodium carbonate-soluble fraction (SSF) [10,11], and 30 mL of 4 M NaOH was used to extract the NaOH-soluble polymers (NaOH mainly solubilizes hemicelluloses, HCF). Finally, the remaining residue was hydrolyzed with 30 mL of 66% (*v*/*v*) H_2_SO_4_ at 37 °C for 1 h. The homogenate was considered to be sulfuric acid-soluble polymers (sulfuric acid removes cellulose fraction, CF). After lyophilization, WSF, CSF, SSF, HCF, and CF were weighed and considered as the contents of the cell wall polysaccharides.

The quantification of WSF, CSF, and SSF were measured with the m-hydroxydiphenyl method using galacturonic acid (GA) as the standard [12]. The results were expressed as g GA kg^−1^ AIR (g kg^−1^) with three replications per sample. The contents of HCF and CF were determined with the anthrone method using glucose (Glu) as the standard [13]. The results were expressed as g Glu kg^−1^ AIR (g kg^−1^) with three replications per sample. Each cell wall’s polysaccharides content was expressed on an AIR weight basis.

### 2.7. Determination of Lignin Content

The lignin content (LC) was analyzed by using a lignin assay kit (Solarbio, Beijing, China). The air-dried (80 °C) pulp used for lignin isolation was ground and passed a 30–50 mesh sieve. Samples (3 mg) were weighed and determined according to the operation procedure in the instruction. The results were expressed as g kg^−1^ on a fresh-weight basis.

### 2.8. Determination of the Molecular Weight of Cell Wall Polysaccharides

The molecular weight (Mw) distribution of cell wall polysaccharides was determined with high-performance gel-permeation chromatography (HPGPC), according to the method of Wen et al. (2011) [14]. Four cell wall polysaccharides extracted in “2.6” were filtered through a 0.22 μm hydrophilic membrane prior to HPGPC analysis. This instrument was equipped with a Shimadzu LC-10A HPLC system and a BRT105-104-102 column (8 mm × 300 mm). The temperature of the differential refraction index detector (RID) and the column were 40 °C during operation. The mobile phase velocity was 0.6 mL min^−1^. For calibration, the standard dextrans (Mp1152, Mp5000, Mp11600, Mp23800, Mp48600, Mp80900, Mp148000, Mp273000, Mp409800, and Mp667800) were applied.

### 2.9. Analysis of the Monosaccharide Compositions of Cell Wall Polysaccharides

The monosaccharide composition of cell wall polysaccharides was determined according to the method of Ren et al. (2020) [15]. Samples were measured by gas chromatography (450-GC, Jinan Saichang Scientific Instrument Co., Ltd., BRUKER, Germany) equipped with a DB-1701 column (30 m × 25 mm × 0.25 μm) and a flame ionization detector (FID). The temperature of the detector was 275 °C during operation. The mobile phase was helium and the flow velocity was 1 mL min^−1^. The scheme for temperature control of the column was as follows: the original temperature of 150 °C was kept for two minutes, then it was raised to 250 °C at a velocity of 10 °C min^−1^ and kept for eight minutes. D-galactose, D-glucose, L-rhamnose, L-arabinose, D-mannose, and D-xylose (Sigma, Seebio Biotech (Shanghai) Co., Ltd., Shanghai, China) were used as standards. 

### 2.10. Determination of Cell-Wall-Modifying Enzymes

The pulp tissue (1.00 g) was ground in a precooled mortar and homogenized with 6 mL of 8.8% NaCl containing 10 g L^−1^ PVPP. After extraction at 4 °C for 1 h, the solution was centrifuged at 12,000× *g* for 20 min. The supernatant was obtained and used to measure enzyme activity.

PG activity was assayed using the method of Yoshida et al. (1984). Decomposition of galacturonic acid (1 mg) per gram sample per hour was considered as one unit of PG activity (U = mg/h/g FW).

Cx activity was determined according to the method of Durbin and Lewis (1988) [16], with slight adjustments. The absorbance was assayed at 540 nm and the inactivated enzyme extract was used as the control group. Catalytic production of 1 μg reducing sugar per gram of tissue per hour was considered as one unit of Cx activity (U = µg/h/g FW).

PME activity was determined using the method of Kumar et al. (2017) [17]. Consumption of 1 μmol of NaOH per one gram of tissue per one hour was considered as one unit of PME activity (U = µmoL/h/g FW).

β-GAL and α-ARF activities were measured using the method of Brummell et al. (2004) [18]. The absorbance was measured at 405 nm. A calibration curve was obtained by using free *p*-nitrophenol (PNP) as the standard. The production of 1 nmol per gram of tissue per minute for PNP was considered as one unit of enzyme activity (U = nmol/min/g FW).

XTH activity was measured using the method of Opazo et al. (2010) [19]. The absorbance (OD value) was determined at 450 nm with a microplate analyzer (MultiskanSky High, Thermo Fisher Scientific (China) Co., Ltd., Shanghai, China) within 15 min. The results were expressed as U/kg.

### 2.11. Statistical Analysis

Considering two factors, storage time and fruit cultivar, the data were processed by two-way analysis of variance (ANOVA) using SPSS software (version 24.0, IBM Corp., Armong, NY, USA). The comparison of means was performed using Duncan’s multiple range test (DMRT). Results were denoted as statistically significant at the *p* < 0.05 level. Correlations among indicators were determined using Pearson’s correlation test.

## 3. Results and Discussion

### 3.1. Fruit size, Soluble Solid Content, and Ascorbic Acid Content in Hami Melons during Different Storage Stages

The fruit length and diameter of ‘XZM 17’ and ‘JHM 25’ decreased at the end of storage (*p* < 0.05), particularly at 15–18 d, while those of ‘CG’ did not have obvious changes (*p* < 0.05) (Table 1). The SSC of the three landraces increased at the beginning of storage (‘CG’ from 8.8% to 9.9%; ‘XZM 17’ from 7.3% to 11.5%; ad ‘JHM 25’ from 9.6% to 10.2%) (*p* < 0.05) (Table 1), which might be due to the fact that the starch in Hami melons was converted to soluble sugar to provide energy for the respiration of the fruit at the early stage of storage [19]. The SSC of ‘CG’, ‘XZM 17’, and ‘JHM 25’ reached peak values on day 3, day 6, and day 3, respectively. With the increases of storage time, especially at the end of storage, the sugar consumed was higher than the converse of starch, resulting in a downward trend in the SSC. At the end of storage, the SSC of ‘CG’, ‘XZM 17’, and ‘JHM 25’ reduced to 8.3%, 9.1%, and 8.4%, respectively. AsA is an essential antioxidant which plays a positive role in maintaining fruit quality [20]. In this study, the content of AsA showed the same changing trend as SSC in the three landraces. The content of AsA of ‘CG’ and ‘JHM 25’ were the highest at the early stage of storage (3 d), which were 0.45 g kg^−1^ and 0.37 g kg^−1^, respectively, while the peak AsA of ‘XZM 17’ appeared at 9 d (0.46 g kg^−1^). Compared to the maximum, the AsA content decreased rapidly in three landraces at the end of storage. Among them, the AsA content of ‘CG’ decreased by 64.44%, higher than that of ‘XZM 17’ (50%) and ‘JHM 25’ (45.95%) (*p* < 0.05).

### 3.2. Analysis of Fruit Firmness, Weight Loss, Decay Incidence, and Respiration Rate

Firmness, a crucial indicator of fruit sensory quality, is highly related to the commercial value and shelf life of postharvest fruit. The decrease of firmness as symptoms of fruit softening is connected with cell wall modification [21,22]. The firmness of Hami melons decreased with the extension of storage time (*p* < 0.05). The firmness of ‘CG’ was always higher than that of ‘XZM 17’ and ‘JHM 25’. On day 18, the firmness of ‘CG’ measured at the distance of 1.0 cm from the epidermis decreased by 50.41% (*p* < 0.05) and that of ‘XZM 17’ and ‘JHM 25’ decreased by 64.49% and 74.94% (*p* < 0.05), respectively (Figure 1A). In general, ‘CG’ maintained a higher firmness at the end of storage, but this was dependent on storage time.

A weight loss of 1.9% was observed in ‘CG’ (Figure 1B), which was lower than that of ‘XZM 17’ (4.7%) and ‘JHM 25’ (9.6%), indicating that ‘CG’ had a lower moisture loss, which could indicate that it can maintain its freshness during storage. Previous studies found that the weight loss of a banana, as well as that of a tomato, elevated during storage, which might be due to the respiration and transpiration of fruit [23,24]. In this study, the decay incidence (Figure 1C) of ‘XZM 17’ and ‘JHM 25’ showed an upward trend with the extension of storage time (Figure 1C). The decay incidence in ‘XZM 17’ (41.62%) and ‘JHM 25’ (53.84%) were the highest on day 18. The increase in weight loss of ‘CG’ was not obvious during the whole storage period, and there was no decay compared with that of ‘XZM 17’ and ‘JHM 25’. The results of decay incidence, firmness, and weight loss indicated that the storage quality of ‘CG’ was obviously better than that of ‘XZM 17’ and ‘JHM 25’. High respirations rate accelerated their metabolic rates, resulting in the consumption of nutrients and water. The respiration rate of ‘JHM 25’ reached its peak in the early stage of storage, and subsequently declined with the extension of storage time, while that of ‘XZM 17’ decreased continuously and reached its minimum on day 18, which was manifested by the increases of weight loss (Figure 1B) and decay incidence (Figure 1C). The respiration rate of ‘CG’ was lower than that of ‘XZM 17’ and ‘JHM 25’ (*p* < 0.05) throughout the storage period, which is consistent with its lower decay incidence and weight loss.

### 3.3. Ultrastructure of Cell Walls in Hami Melons

Complete cell walls, uniformity thickness, and clear middle lamellae were observed in the three landraces on day 0 (Figure 1E,G,I). Moreover, the primary cell walls of Hami melon flesh were tightly attached to both sides of the middle lamellae, showing a bright-dark-bright partition structure. On day 18, the cell wall structure of the three Hami melon landraces became loose, with apparent wrinkles and non-uniform thickness. In addition, the middle layer started to be depolymerized, especially in ‘XZM 17’ and ‘JHM 25’ (Figure 1F,H). The cell walls of ‘CG’ (Figure 1J) were only slightly degraded at the end of storage compared with that of ‘JHM 25’ and ‘XZM 17’. The above results indicated that at the initial stage of storage, the cell wall structure was relatively intact, with only a few stray microfibrils, and there were no swelling deformation of the chlorophylls and vacuole therein, and no apparent plasmolysis. At the end of storage, the cell wall structure of the three Hami melon varieties was loose and obviously wrinkled, and the thickness was not uniform. This indicated cell wall degradation in the three Hami melon landraces during storage (*p* < 0.05). Meanwhile, there were differences in cell wall degradation among the three landraces (*p* < 0.05).

### 3.4. Cell Wall Material Content and the Change of Cell Wall Composition during Storage

On day 0, the content of CWM of ‘CG’ was higher than that of ‘XZM 17’ and ‘JHM 25’ (*p* < 0.05). The content of CWM decreased with the extension of storage time (*p* < 0.05) (Table 1), which might be due to Hami melon being a climacteric fruit. On day 18, the CWM content decreased to the lowest value. Among them, the CWM content of ‘JHM 25’ (0.0237 g kg^−1^) was lower than that of ‘CG’ (0.0477 g kg^−1^) and ‘XZM 17’ (0.0366 g kg^−1^) (*p* < 0.05), which was consistent with the uppermost decay incidence and weight loss in ‘JHM 25’ during storage (Figure 1B,C). Wang et al. (2017) found that the CWM content of untreated blueberry fruit decreased more than that of the γ-irradiated group in storage, which might be caused by the degradation of cell wall components. In our study, the CWM content of ‘XZM 17’ and ‘JHM 25’ continuously declined during storage, which further confirmed that the cell wall components of Hami melon were degraded. Within cell wall components, WSF, CSF, and SSF are three kinds of pectin polysaccharides, which are crucial components of cell walls. In addition, their changes could induce fruit ripening and softening [2,25,26]. In this study, WSF and CSF, representing loosely and ionically bound pectins, respectively, rose with the extension of storage time (*p* < 0.05), whereas SSF, representing a tightly bound polymer, decreased (*p* < 0.05) (Table 1). The content of WSF was higher than that of CSF and SSF at end stages of storage (*p* < 0.05). At the same time, ‘CG’ had the highest content of WSF and SSF, followed by ‘JHM 25’ and ‘XZM 17’. It was worth noting that there was a negative correlation (‘CG’, r = −0.64; ‘JHM 25’, r = −0.71; ‘XZM 17’, r = −0.56) between CSF and SSF throughout the storage period, indicating that SSF was derived from CSF. The replacement of pectin molecules indicated the changes in pectin structure during fruit ripening and softening [4]. In addition, the content of HCF and CF were lower than that of SSF and slowly declined with the extension of storage time. The HCF content of ‘CG’ was higher than that of ‘XZM 17’ and ‘JHM 25’ at 0 d (*p* < 0.05), where the same situation also appears in CF. Cellulose and hemicellulose play an important role in supporting the integrity of cell wall structure, and their highest content in ‘CG’ corresponded to the highest fruit firmness (Figure 1A). Lignin content can also be used to characterize the changes in cell wall composition. The content of lignin reached its peak on day 6 and no difference was found on day 0–3 and 9–15 (*p* > 0.05). Among them, the ‘CG’ landrace has the greatest content of lignin (1.5 g/kg) on day 6. Taken together, the three Hami melon landraces had various flesh characteristics and cell wall compositions, which were closely connected with fruit softening.

### 3.5. Analysis of the Molecular Weight of Cell Wall Polysaccharides

The change of cell wall polysaccharides was considered to be an important reason for the decrease of fruit firmness and the change of cell wall structure during storage [27,28]. Figure 2 exhibits the molecular weight distribution curve of WSF (Figure 2A), CSF (Figure 2B), SSF (Figure 2C), and HCF (Figure 2D) at different stages of storage. The molecular weight was estimated on the basis of the calibration with standard dextrans derived from linear regression (logMw = −0.1889 t + 12.007, r^2^ = 0.9943). The curve of WSF had multiple peaks (Peak 1–4) and the retention time was mainly 30–55 min for the three Hami melon landraces, according to the HPGPC system. The peak on day 0 was broader than that on day 18. The Mw in different molecular weight regions all decreased with the extension of storage time. In addition, new macromolecules such as ‘JHM 25’ (274517 Da) and ‘XZM 17’ (682833 Da) appeared in the Mw curve on day 18, which might be due to the conversion of insoluble pectin to water-soluble pectin during storage. These results could well explain the depolymerization of macromolecular polymers and the production of new macromolecular polymers during the softening process [29]. There were also two kinds of molecular weight regions for CSF. The area of high-molecular-weight polymer in ‘XZM 17’ reduced, while that of the low-molecular-weight pectin increased with the extension of storage time. Pectin isolated from Hami melon fruit at the end of storage showed a lower Mw, which might be due to the degradation of pectin by enzymes such as PME and PG [30]. At the same time, the contrary results of ‘CG’ and ‘JHM25’ might be caused by the interspecific differences. SSF had the widest distribution (25–55 min) and the highest molecular weight (peak 1 > 1000 kDa) in pectin among the three landraces. The peak area of pectin macromolecule polymer of the three landraces decreased on day 18, and that of ‘JHM 25’ decreased the most, which might be due to the depolymerization of the long pectin chains of SSF during storage [30]. The retention time of HCF molecular weight was mainly in 35–50 min and the variation of the peak was similar to that of WSF, CSF, and SSF. Especially, the peak number of ‘CG’ decreased obviously (from 4 to 2) at the end of storage. Similarly, the increase in the number of peaks of ‘XZM 17’ from 2 to 4 also indicated great change during storage. The molecular weight distribution of WSF, CSF, SSF, and HCF all indicated cell wall polysaccharide degradation (Figure 2). Combined with the changes in firmness and pectin content, the results indicated that the softening process of fruit was greatly associated with the variation of pectin components of primary cell wall polysaccharides. Chen et al. (2021) held that the cell wall polysaccharides disassembly induced longan fruit softening. Wang et al. (2017) [31] showed that the number of absorption peaks of high molecular weight polymers in WSF, CSF, and SSF in blueberry fruit continuously declined, indicating that the high molecular weight polymers were degraded into low molecular weight polymers. The variation of molecular weight further confirmed the depolymerization and degradation of pectin molecules during the softening of Hami melons, which might cause the difference in storage quality of different Hami melon landraces.

### 3.6. Monosaccharides Compositions of Cell Wall Polysaccharides

The changes of the monosaccharide composition of cell wall polysaccharides have a great influence on the storage quality of fruit after harvest [32,33,34]. As shown in Figure 3, glucose, galactose, and arabinose were the main neutral monosaccharides in WSF (Figure 3A,E,I), CSF (Figure 3B,F,J), and SSF (Figure 3D,H,L). Glucose had the highest content in WSF of different Hami melon landraces, followed by galactose and arabinose. Furthermore, there was also a small quantity of mannose and rhamnose in each cell wall polysaccharide, along with a certain amount of xylose in HCF (Figure 3D,H,L). The glucose content was the highest in WSF, while that in CSF and SSF decreased. Particularly, the content of glucose generally decreased with storage, indicating the destroyed cell wall framework and fruit softening. Arabinose and galactose, the neutral sugar side chains of rhamnogalacturonan I, can be covalently linked to hemicellulose and cellulose, and the removal and rearrangement of them are highly related to the cell wall porosity and strength [35]. Hence, the variation of content in arabinose and galactose in large amounts indicated the rhamnogalacturonan I in CSF and SSF. Different from WSF, CSF, and SSF, glucose and xylose appeared to be dominant in HCF. With a prolonged storage time, the relative content of xylose in HCF tended to increase gradually as a whole, which indicated that the hemicellulose fraction contained either xylogalacturonan or xylose. In summary, the high glucose, galactose, and arabinose contents in WSF, CSF, and SSF were more likely released from the pectin polysaccharides of the primary cell wall. Their changes in relative contents during softening of the Hami melons were in fact the best illustration for the degradation of cell wall pectin polysaccharides. The monosaccharides composition of HCF displayed high glucose and xylose contents, suggesting that xyloglucan might be present in the hemicellulose. Other minor amounts of monosaccharides (including mannose and rhamnose), probably due to the presence of polygalacturonic acid, were tightly bonded to the cell wall and only strict extraction conditions could extract them. The reduction content of glucose in HCF was also an indication of the destroyed cell wall framework and fruit softening. The variation in the relative content further confirmed the degradation of cell wall pectin polysaccharides during softening of the Hami melons.

### 3.7. Cell-Wall-Degrading Enzyme Activities

During fruit softening, the changes of cell wall materials mainly stemmed from the variation in the cell wall modifying enzyme activities. PG, Cx, PME, XTH, α-ARF, and β-GAL were cell-wall-degrading enzymes involved in fruit softening [2,3,36,37]. The activities concerning Cx, PG, α-ARF, or β-GAL regularly multiplied with the development of storage time. The opposite trend used to be found in the activity of XTH and PME (Figure 4A–F). These results indicated that the four enzymes (Cx, PG, α-ARF, and β-GAL) played a major role in the decreasing fruit firmness. The Cx activity in ‘JHM 25’ was sharply enhanced during storage (Figure 4A), higher than that of ‘CG’ and ‘XZM 17’ (*p* < 0.05). The Cx activity of ‘CG’ increased rapidly at the beginning of storage (on day 0–5), but that of ‘XZM 17’ increased in the middle and late stages (on day 6–18). PG is the most important pectin modifying enzyme during the softening of climacteric fruit [38]. Before activating PG, PME demethylation of pectin chains is necessary, which produces a suitable substrate for PG [3]. In this study, the PG activity increased gradually and peaked at the late stage of storage (on day 15–18) (Figure 4D), indicating that PG might play a weak role at the early stage of fruit softening. At the same time, the PME activity of the three landraces decreased continuously during storage (Figure 4C), which was identical to the trend of firmness (Figure 1A). A high PME activity at the beginning of storage could help the conversion of methylesterified polyuronide to demethylesterified polyuronide, but a low PG activity had no ability to deal with so many substrates, eventually leading to the formation of a hard texture of cell wall structure. A low PME activity and a high PG activity are bound to lead to the fruit subsequently softening [3].

The activity of α-ARF and β-GAL have an impact on cell wall structure. Further, they can help accelerate the dissolution of pectin by removing the monosaccharides, such as arabinose and galactose, from the side chains of pectin, thus increasing the chances for other cell-wall-modifying enzymes such as PG to contact with reaction substrates [4]. Similarly, the activities of α-ARF and β-GAL in Hami melons additionally extended with the extension of storage time (Figure 4E,F), which was somewhat different from the research results of Chea et al. (2019). They concluded that the activity of α-ARF and β-GAL in highbush blueberries decreased during fruit ripening, which might be ascribed to the differences in strain. The degradation of hemicellulose occurs during fruit softening, and xylan is the most important cell wall hemicellulose. Therefore, the degradation of xylan is also regarded to be a crucial feature of fruit softening. The XTH enzyme is believed to play an important role in fruit ripening. It loosens the cell wall to facilitate the further modification of other cell wall metabolity-related enzymes and the decomposition of xyloglucan [39]. In the present study, the activity of XTH was higher at the initial stage of storage; however, it decreased with the extension of storage time (*p* < 0.05) (Figure 4B). Similar to the results mentioned above, Miedes et al. (2009) found that the activity of XTH decreased during tomato ripening, which may be a reason for fruit softening. The above results indicated that PME and XTH maintained a high activity at the beginning of storage, and the softening occurring at the late stage may stem from the increased activities of PG, β-Gal, α-ARF, and Cx. Our findings indicated that related metabolic enzymes of the cell wall were involved in fruit softening of the Hami melons at different stages of storage. Similar findings were also found in the ‘Bluecrop’ highbush blueberry [4] and *Annona squamosal* [3].

### 3.8. Correlation Analysis among Fruit Quality Characteristics and Cell Wall Components, as well as Cell-Wall-Modifying Enzymes in the Three Hami Melon Landraces

Correlation analysis (Figure 5A–C) showed that the firmness of the Hami melon fruit had a negative correlation with WSF and a positive correlation with CWM and CF. The results indicated that CWM, WSF, and CF had a great impact on fruit softening. Therefore, the softening degree of Hami melon fruit can be predicted through determining the changes in the content of cell wall components. Wang et al. (2021) found that the cell wall degradation during the softening of kiwifruit fruit was highly related to the changes in the composition of cell wall. Findings of the same kind were also obtained by previous studies [21,22,29,40,41,42,43].

The relationships between fruit softening and cell wall metabolic enzymes were consistent for different species [2]. For example, the softening of the ‘Jingbaili’ pear was closely related to the changes of β-GAL and α-ARF activities, and a similar finding was also found in the ‘La France’ pear, indicating that the softening mechanisms of the ‘Jingbaili’ and ‘La France’ pear fruits were similar [2,44]. In our research, correlation analysis (Figure 5) showed that the firmness of the three Hami melon landraces had a negative correlation with PG, β-Gal, α-ARF, and Cx, and a positive correlation with PME and XTH. Our findings suggest a two-phase response that early softening in Hami melon fruits is caused, at least in part, by higher activities of XTH and PME, while late softening stems from the increased activities of Cx, PG, α-ARF, and β-Gal. CWM also showed a similar correlation with the cell wall metabolic enzymes. Undeniably, fruit softening itself was a complex process and the involvement of other cell-wall-modifying enzymes in the fruit softening of postharvest Hami melons was also possible, which deserves deeper studied in the future.

## 4. Conclusions

Compared with ‘XZM 17’ and ‘JHM 25’, ‘CG’ had a higher storage quality at the end of storage. The pulp of ‘CG’ maintained a higher firmness and a higher content of cell wall components, as well as a more intact cell wall structure with only a few free microfibrils. In addition, the softening mechanisms of the three landraces were similar from the perspectives of molecular weight and monosaccharide composition in different pectin polysaccharides and cell wall metabolic enzymes. The higher storage quality of ‘CG’ may be attributed to the higher content of cell wall components than the other two landraces, which should be given full consideration for future research to solve the problem of fruit quality deterioration of Hami melons under different storage conditions. On the whole, the softening of Hami melon fruit is correlated with the reduction in CWM, structural changes of pectin polysaccharides, and the role of cell wall metabolic enzymes. This experiment provides a theoretical basis for future study on the inhibition of fruit softening, shelf life extension, and fruit quality improvement of postharvest Hami melons.

## Figures and Tables

**Figure 1 foods-11-00841-f001:**
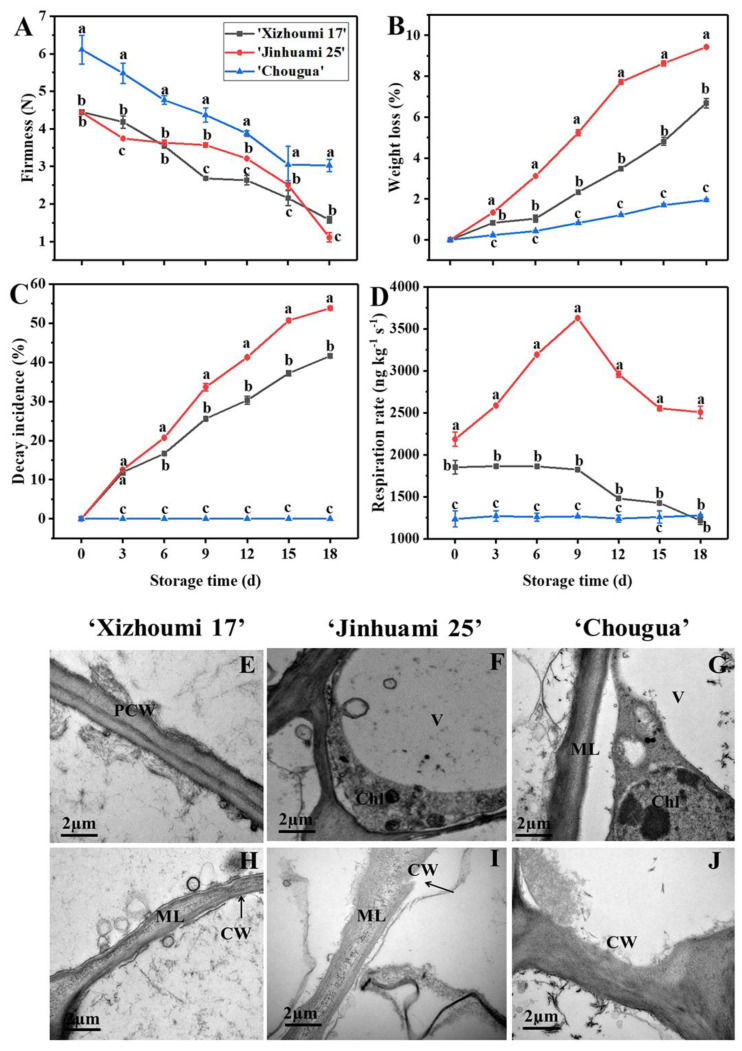
Change of firmness (**A**), weight loss (**B**), decay incidence (**C**), respiration rate (**D**), and ultrastructure (**E**–**J**) of the cell walls of Hami melons during storage at 20 °C for 18 days. Note: the data presented are the means of three replicates from three experiments; vertical bars represent the standard errors of the means; values followed by different superscripts (a-c) are significantly different (*p* < 0.05) in the same sampling time; 0, 3, 6, 9, 12, 15, and 18 represent the days after Hami melon harvest; (**E**,**H**) represent ‘Xizhoumi 17’ at 0 d and 18 d of storage, respectively; (**F**,**I**) represent ‘Jinhuami 25’ at 0 d and 18 d of storage, respectively; (**G**,**J**) represent ‘Chougua’ at 0 d and 18 d of storage, respectively; CW, V, Chl, ML, and PCW denote cell wall, vacuole, chloroplast, middle lamella, and primary cell wall, separately. Appendix A. The three Hami melon landraces (**A**: ‘Xizhoumi 17’; **B**: ‘Jinhuami 25’; and **C**: ‘Chougua’) from Xinjiang Province, China, used in this study.

**Figure 2 foods-11-00841-f002:**
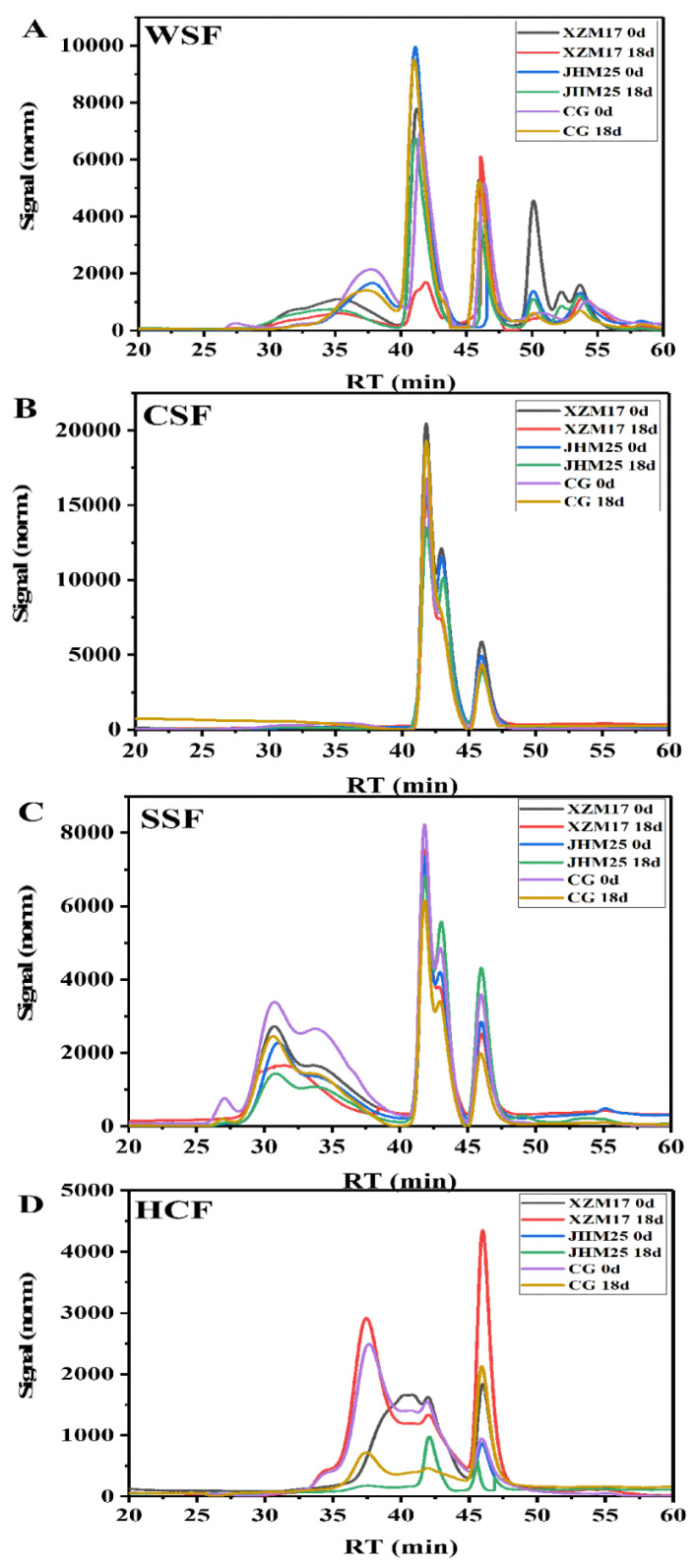
Molecular weight distribution curve in WSF (**A**), CSF (**B**), SSF (**C**) and HCF (**D**) of the three Hami melon landraces (‘XZM 17’, ‘JHM 25’ and ‘CG’) and during storage at 20 °C for 18 d. Note: ‘XZM 17’, Xizhoumi 17; ‘JHM 25’, Jinhuami 25; ‘CG’, Chougua; WSF, water-soluble fraction; CSF, chelate-soluble fraction; SSF, sodium carbonate-soluble fraction; HCF, hemicelluloses fraction; 0 and 18 represent the days after Hami melon harvest; RT: retention time; norm: normal.

**Figure 3 foods-11-00841-f003:**
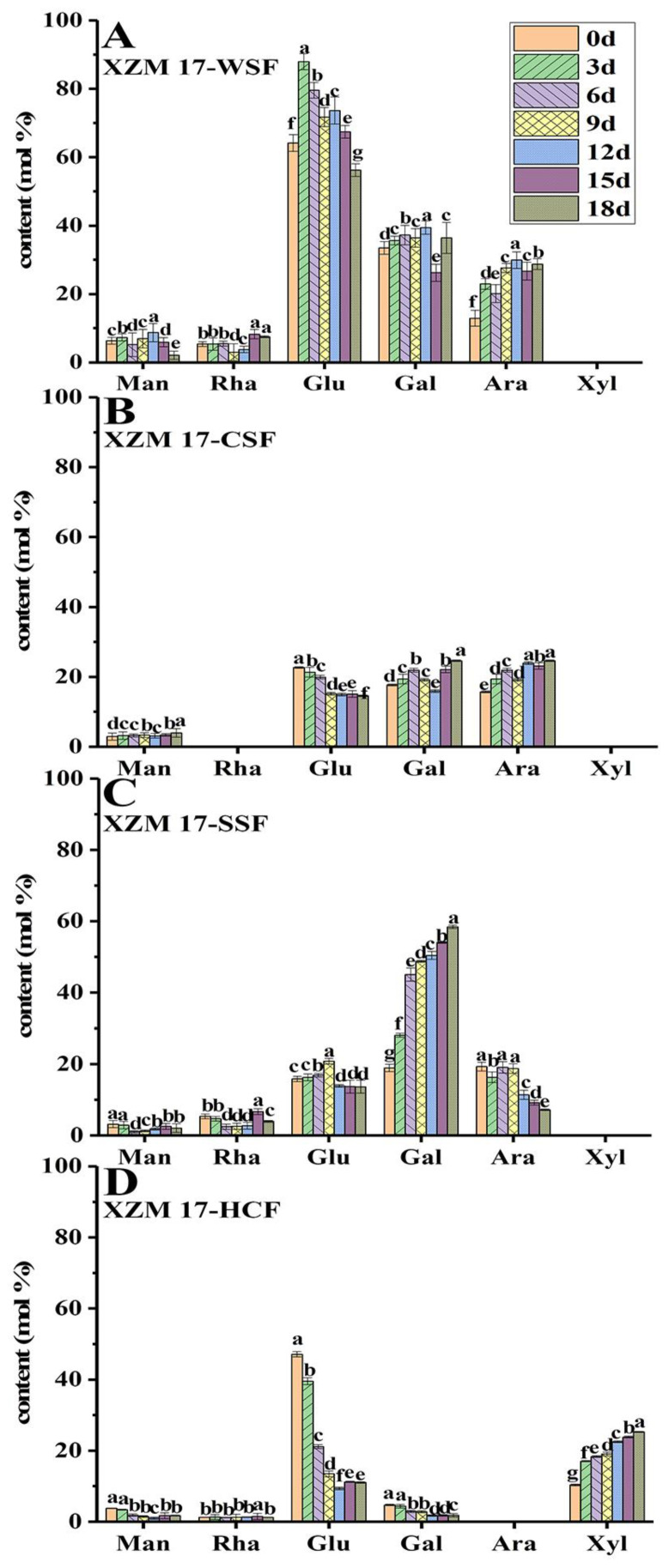
Monosaccharide composition of pectin polysaccharides and hemicelluoses in Hami melons during storage. Note: ‘XZM 17’, ‘Xizhoumi 17’; ‘JHM 25’, ‘Jinhuami 25’; ‘CG’, ‘Chougua’; WSF, water-soluble fraction; CSF, chelate-soluble fraction; SSF, sodium carbonate-soluble fraction; HCF, hemicelluloses fraction; (**A**–**D**) WSF, CSF, SSF, and HCF of ‘XZM 17’; (**E**–**H**) WSF, CSF, SSF, and HCF of ‘JHM 25’; (**I**–**L**) WSF, CSF, SSF, and HCF of ‘CG’; 0, 3, 6, 9, 12, 15, and 18 represent the days after Hami melon harvest; values are the means of three replicates (*n* = 3) ± SD (standard deviation); vertical bars represent the standard deviation (*p* < 0.05); different letters indicate significant differences during storage; x-coordinates represent mannose (Man), rhamnose (Rha), glucose (Glu), galactose (Gal), arabinose (Ara), and xylose (Xyl) in sequence.

**Figure 4 foods-11-00841-f004:**
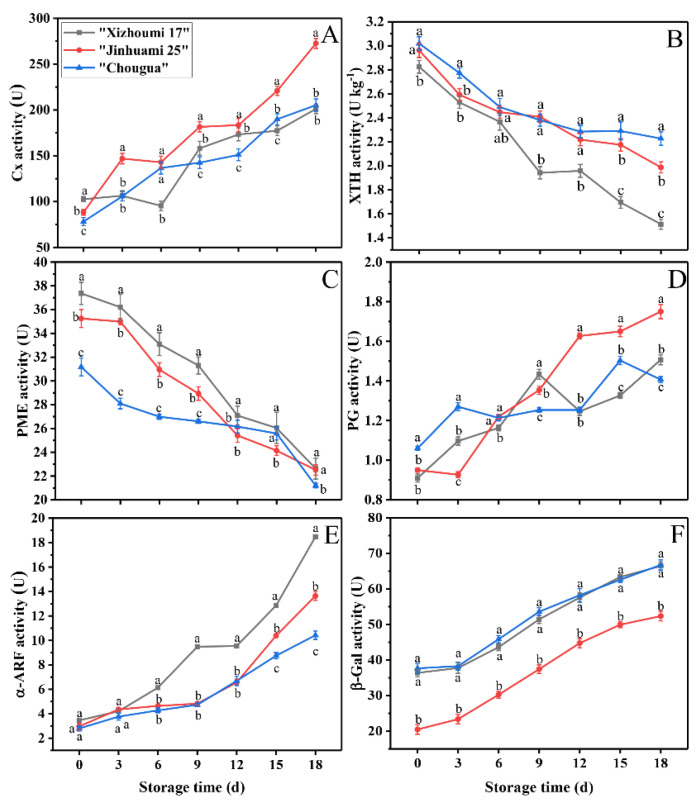
Change of relating enzyme activities with fruit softening: ((**A**) Cx, (**B**) XTH, (**C**) PME, (**D**) PG, (**E**) α−ARF, and (**F**) β−GAL) in Hami melons during 18 days of storage. Note: Cx, cellulase; XTH, xyloglucan endo−transglycosylase/hydrolase; PME, pectin methylesterase; PG, polygalacturonase; α−ARF, α−arabinofuranosidase; β−GAL, β−galactosidase; values are the means of three replicates (*n* = 3) ± SD (standard deviation); vertical bars represent the standard deviation (*p* < 0.05); different letters indicate significant differences during the same sampling time; 0, 3, 6, 9, 12, 15, and 18 represent the days after Hami melon harvest.

**Figure 5 foods-11-00841-f005:**
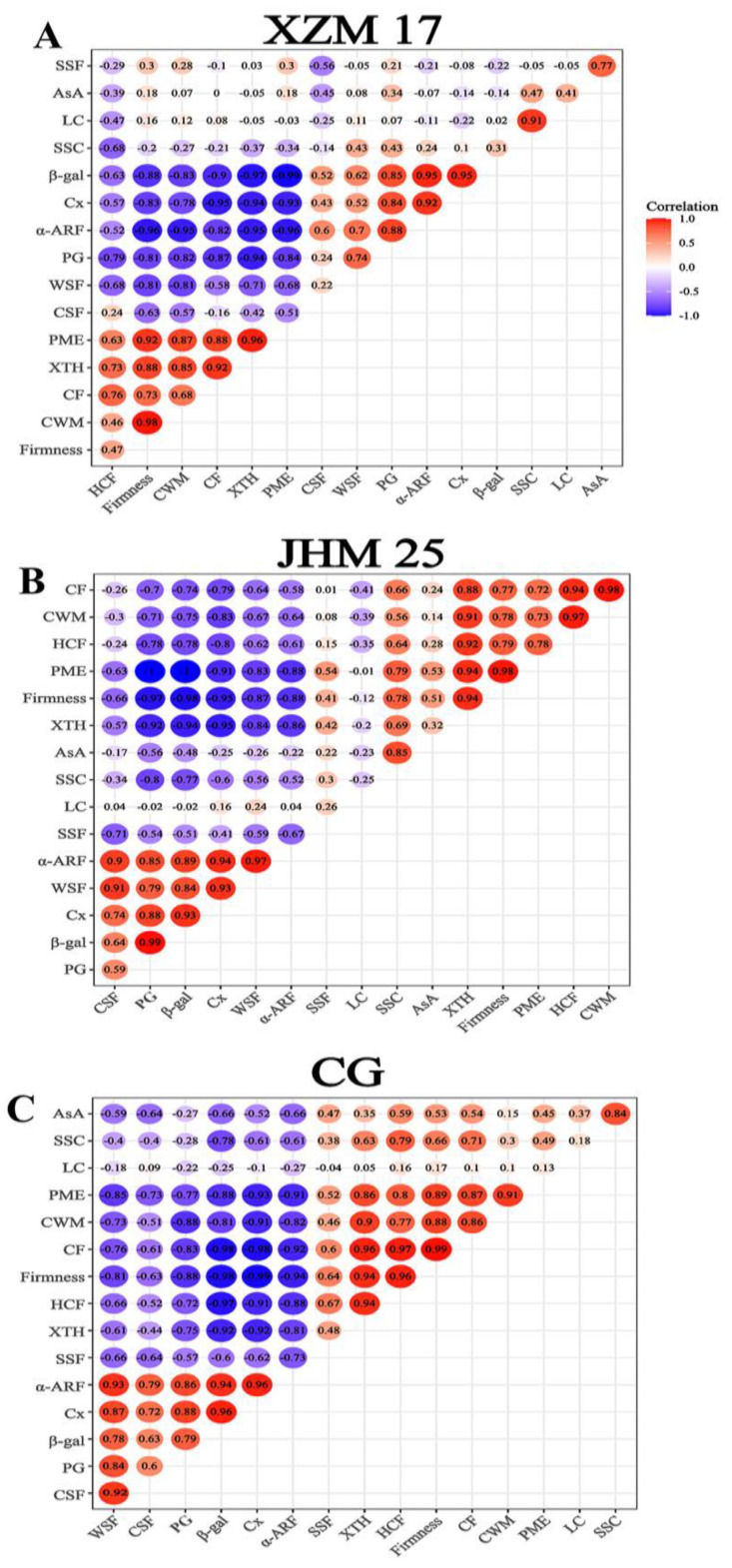
The Pearson correlation between physicochemical indexes, cell wall components, and cell-wall-modifying enzymes of the three Hami melon landraces ((**A**) ‘XZM 17’; (**B**) ‘JHM 25’; and (**C**) ‘CG’) during storage. Note: ‘XZM 17’, Xizhoumi 17; ‘JHM 25’, Jinhuami 25; ‘CG’, Chougua; SSC, soluble solid content; CWM, cell wall material; WSF, water-soluble fraction; CSF, chelate-soluble fraction; SSF, sodium carbonate-soluble fraction; HCF, hemicelluloses fraction; CF, cellulose fraction; LC, lignin content; Cx, cellulase; XTH, xyloglucan endo-transglycosylase/hydrolase; PME, pectin methylesterase; PG, polygalacturonase; α-ARF, α-arabinofuranosidase; β-GAL, β-galactosidase. The dark color represents a strong correlation, the light color represents a weak correlation, red represents a positive correlation, and blue represents a negative correlation.

**Table 1 foods-11-00841-t001:** Quality attributes, CWM content, and cell wall polysaccharides composition in Hami melon fruits during 18 days of storage at 20 °C.

Storage Time (d)	Length (cm)	Diameter (cm)	SSC (%)	AsA (g kg^−1^)	CWM (g kg^−^^1^)	WSF (g kg^−^^1^)	CSF (g kg^−^^1^)	SSF (g kg^−^^1^)	HCF (g kg^−^^1^)	CF (g kg^−^^1^)	LC (g kg^−^^1^)
0 (‘XZM 17’)	19.20 ± 0.5 ^a^	13.76 ± 0.4 ^a^	7.3 ± 0.3 ^c^	0.19 ± 0.09 ^f^	0.0505 ± 0 ^a^	1.4 ± 0 ^f^	6.3 ± 0.4 ^b^	5.2 ± 0.7 ^cd^	0.8 ± 0 ^a^	0.8 ± 0 ^a^	0.6 ± 0 ^c^
3	19.19 ± 0.2 ^a^	13.74 ± 0.3 ^a^	7.7 ± 0.4 ^c^	0.25 ± 0.07 ^c^	0.0465 ± 0 ^b^	17.8 ± 2.3 ^e^	1.8 ± 0.6 ^cd^	16.2 ± 0.2 ^b^	0.5 ± 0 ^b^	0.6 ± 0 ^a^	0.6 ± 0 ^c^
6	19.11 ± 0.1 ^a^	13.70 ± 0.5 ^a^	11.5 ± 0.3 ^a^	0.33 ± 0.04 ^b^	0.0420 ± 0 ^c^	32.5 ± 1.4 ^c^	3.5 ± 0.2 ^c^	17.9 ± 0.8 ^cd^	0.4 ± 0 ^b^	0.7 ± 0 ^a^	1.1 ± 0 ^a^
9	18.98 ± 0.2 ^a^	13.56 ± 0.2 ^a^	9.7 ± 0.4 ^b^	0.46 ± 0.06 ^a^	0.0418 ± 0 ^d^	29.4 ± 4.3 ^cd^	0.9 ± 0 ^d^	24.4 ± 2.0 ^a^	0.3 ± 0 ^b^	0.3 ± 0 ^b^	0.8 ± 0 ^b^
12	18.93 ± 0.3 ^a^	13.52 ± 0.4 ^b^	9.5 ± 0.2 ^b^	0.17 ± 0.07 ^g^	0.0413 ± 0 ^e^	25.0 ± 1.8 ^d^	1.3 ± 0 ^cd^	5.7 ± 0.5 ^cd^	0.3 ± 0 ^b^	0.2 ± 0 ^b^	0.8 ± 0 ^b^
15	18.89 ± 0.5 ^b^	13.46 ± 0.3 ^b^	9.2 ± 0.4 ^b^	0.21 ± 0.08 ^e^	0.0383 ± 0 ^f^	71.4 ± 1.5 ^b^	10.6 ± 0.1 ^a^	7.8 ± 1.0 ^c^	0.4 ± 0 ^b^	0.3 ± 0 ^b^	0.8 ± 0 ^b^
18	18.85 ± 0.4 ^b^	13.43 ± 0.2 ^b^	9.1 ± 0.3 ^b^	0.23 ± 0.05 ^d^	0.0366 ± 0 ^g^	106.2 ± 3.9 ^a^	10.1 ± 0.5 ^a^	4.2 ± 0.5 ^cd^	0.4 ± 0 ^b^	0.2 ± 0 ^b^	0.6 ± 0 ^c^
0 (‘JHM 25’)	25.63 ± 0.6 ^a^	13.56 ± 0.5 ^a^	9.6 ± 0.3 ^a^	0.22 ± 0.04 ^b^	0.0457 ± 0 ^a^	16.8 ± 0.9 ^e^	5.8 ± 0.3 ^c^	10.0 ± 0.2 ^c^	0.7 ± 0 ^a^	0.9 ± 0 ^a^	0.5 ± 0 ^b^
3	25.61 ± 0.4 ^a^	13.55 ± 0.2 ^a^	10.2 ± 0.3 ^a^	0.37 ± 0.03 ^a^	0.0395 ± 0 ^b^	36.1 ± 0.8 ^c^	3.1 ± 0.1 ^cd^	17.7 ± 0.3 ^b^	0.4 ± 0 ^b^	0.4 ± 0 ^b^	0.6 ± 0 ^b^
6	25.56 ± 0.6 ^a^	13.51 ± 0.3 ^a^	8.3 ± 0.4 ^b^	0.17 ± 0.07 ^e^	0.0389 ± 0 ^c^	37.9 ± 2.4 ^c^	3.9 ± 0 ^c^	18.4 ± 1.1 ^c^	0.3 ± 0 ^b^	0.3 ± 0 ^b^	0.8 ± 0 ^a^
9	25.50 ± 0.3 ^a^	13.47 ± 0.4 ^a^	8.2 ± 0.4 ^b^	0.17 ± 0.06 ^e^	0.0372 ± 0 ^d^	27.3 ± 1.8 ^d^	2.9 ± 0.2 ^d^	24.0 ± 1.8 ^a^	0.3 ± 0 ^b^	0.2 ± 0 ^b^	0.7 ± 0 ^a^
12	25.47 ± 0.2 ^a^	13.43 ± 0.5 ^a^	8.5 ± 0.3 ^b^	0.19 ± 0.02 ^d^	0.0370 ± 0 ^d^	34.2 ± 0.8 ^c^	2.1 ± 0 ^d^	9.2 ± 0.8 ^c^	0.2 ± 0 ^b^	0.3 ± 0 ^b^	0.5 ± 0 ^b^
15	25.44 ± 0.5 ^a^	13.40 ± 0.5 ^a^	7.9 ± 0.2 ^bc^	0.20 ± 0.03 ^c^	0.0343 ± 0 ^e^	106.7 ± 4.7 ^b^	12.5 ± 1.6 ^b^	3.9 ± 0 ^d^	0.3 ± 0 ^b^	0.2 ± 0 ^b^	0.5 ± 0 ^b^
18	25.38 ± 0.3 ^b^	13.34 ± 0.2 ^b^	8.4 ± 0.3 ^b^	0.20 ± 0.05 ^c^	0.0237 ± 0 ^f^	142.5 ± 6.7 ^a^	18.6 ± 0.1 ^a^	3.4 ± 0 ^d^	0.2 ± 0 ^b^	0.2 ± 0 ^b^	0.7 ± 0 ^a^
0 (‘CG’)	17.56 ± 0.1 ^a^	12.02 ± 0.5 ^a^	8.8 ± 0.2 ^a^	0.23 ± 0.01 ^f^	0.0528 ± 0 ^a^	23.3 ± 2.4 ^d^	7.0 ± 0.9 ^cd^	19.2 ± 0.6 ^b^	0.9 ± 0 ^a^	1.0 ± 0 ^a^	0.7 ± 0 ^b^
3	17.56 ± 0.2 ^a^	12.01 ± 0.3 ^a^	9.9 ± 0.3 ^a^	0.45 ± 0.03 ^a^	0.0499 ± 0 ^b^	54.5 ± 3.8 ^b^	4.9 ± 0.1 ^d^	19.8 ± 0.1 ^b^	0.9 ± 0 ^a^	0.9 ± 0 ^a^	0.6 ± 0 ^b^
6	17.52 ± 0.1 ^a^	12.01 ± 0.4 ^a^	9.0 ± 0.1 a	0.38 ± 0.04 ^b^	0.0483 ± 0 ^c^	54.6 ± 3.0 ^b^	9.1 ± 1.0 ^c^	23.0 ± 0.3 ^c^	0.6 ± 0 ^b^	0.6 ± 0 ^b^	1.5 ± 0 ^a^
9	17.49 ± 0.3 ^a^	12.00 ± 0.3 ^a^	8.5 ± 0.5 ^ab^	0.31 ± 0.04 ^c^	0.0478 ± 0 ^d^	46.6 ± 9.1 ^c^	4.5 ± 0.3 ^d^	28.4 ± 0.6 ^a^	0.5 ± 0 ^b^	0.5 ± 0 ^bc^	0.6 ± 0 ^b^
12	17.49 ± 0.1 ^a^	11.99 ± 0.5 ^a^	8.3 ± 0.4 ^b^	0.27 ± 0.02 ^d^	0.0478 ± 0 ^d^	49.2 ± 3.5 ^bc^	5.3 ± 0.1 ^d^	8.6 ± 0.9 ^d^	0.2 ± 0 ^c^	0.4 ± 0 ^c^	0.6 ± 0 ^b^
15	17.47 ± 0.1 ^a^	11.98 ± 0.3 ^a^	8.5 ± 0.3 ^ab^	0.25 ± 0.08 ^e^	0.0478 ± 0 ^d^	51.8 ± 0.2 ^b^	11.4 ± 0.9 ^b^	6.9 ± 0.7 ^de^	0.2 ± 0 ^c^	0.2 ± 0 ^d^	0.7 ± 0 ^b^
18	17.46 ± 0.2 ^a^	11.96 ± 0.2 ^a^	8.3 ± 0.3 ^b^	0.16 ± 0.05 ^g^	0.0477 ± 0 ^d^	70.6 ± 1.0 ^a^	16.5 ± 0.7 ^a^	8.5 ± 0.5 ^d^	0.2 ± 0 ^c^	0.2 ± 0 ^d^	0.6 ± 0 ^b^

Note: the values are the means of three replicates ± SD (standard deviation); means with different superscripts letters in the same column are statistically different (*p* < 0.05); ‘XZM 17’, Xizhoumi 17; ‘JHM 25’, Jinhuami 25; ‘CG’, Chougua; SSC, soluble solid content; ASA, ascorbic acid; CWM, cell wall material; WSF, water-soluble fraction; CSF, chelate-soluble fraction; SSF, sodium carbonate-soluble fraction; HCF, hemicelluloses fraction; CF, cellulose fraction; LC, lignin content. CWM and LC are expressed on a fresh-weight basis; WSF, CSF, SSF, HCF, and CF contents are expressed on an AIR weight basis; 0, 3, 6, 9, 12, 15, and 18 represent the days after Hami melon harvest.

## Data Availability

Data sharing not applicable. No new data were created or analyzed in this study. Data sharing is not applicable to this article.

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
