# Peer review of "The Role of Cell Wall Polysaccharides Disassembly and Enzyme Activity Changes in the Softening Process of Hami Melon (Cucumis melo L.)"

_foods, 2022, doi:10.3390/foods11060841_

Round 1
Reviewer 1 Report
This paper presented for the review is dedicated to the investigation of changes in composition and ultrastructure of cell walls, firmness and activities of enzyme involving in softening for three landraces of Hami melon, in order to explore the differences in theirs storability.
The manuscript includes extensive research, a lot of analysis is involved, the obtained results are adequately processed, and considered thoroughly.
Only few specific comments are listed:
In the title mention the enzymes that also play a role in softening process of Hami melon.
The general objection is that many abbreviations have been included through the results and discussion, so it is difficult to read the paper carefully and with understanding.
Line 102, Give information about the aparature for TSEM
Line 118, What type of lyophilization device was used?
Line 293, Discuss and compare the content of HCF and CF for all three Hami melon landraces studied
Line 296, Emphasize that for 'CG' landrace is the greatest content of lignin, 1.5g/kg on day 6
Line 460-468 Try to explain why the firmness had a negative correlation with some cell wall modifying enzymes, and a positive correlation with some others.
Reviewer 2 Report
Overall, the manuscript titled "The role of cell wall polysaccharides disassembly and enzyme activity changes in the softening process of Hami melon" is interesting to the broader audience and the authors have used appropriate techniques for the study. Nonetheless, there are several comments, questions and suggestions for the others to improve the manuscript.
- Title: To avoid any confusion that the “Hami” melon is not a different species, the author should provide scientific name and mention “Hami” as cultivar
- Line 16-18: “Such as the depolymerization of macromolecular polymers, the production of new macromolecular polymers and the change of pectic monosaccharides (glucose, galactose and arabinose) content during the softening process of Hami melon”. The description in the sentence does not make sense. Of course, all polymers are macromolecules. Glucose, galactose and arabinose are not pectic monosaccharides as pectin is mainly composed of galacturonic acid monomers which is not galactose even though it is derived from glucose.
- Line 19-21: It should be “cellulase” not cellulose. As a general comment, there are many typos and grammar issues which the author should pay attention while revising the manuscript.
- Experimental section: Line 67-86: I would recommend authors combine sections 2.1 and 2.2 and write methods they used to in measurements (diameter, length) of fruit samples first and then give the numbers. Currently, it is other way around. Also in section 2.2, give full form of SSC and AsA in the heading.
- Section 2.4, Line 104-109: The authors should briefly give the pretreatment method used by Fan et al. (2019) not just the reference. Is there any reason for the authors to use particular reagents to dye the flesh they observed under SEM. How the water from the flesh was removed before observation under SEM?
- Section 2.8, Line 152-160: The authors should briefly give methods used Wen et al. (2011) not just the reference. A reader would want to know at least briefly how the pulp was processed before analysis without the need for searching for the reference.
- Table 1. The authors have provided full forms some of the acronyms used in the table in the foot note (e.g. SSC, soluble solid content; ASA, ascorbic acid) but full forms of other acronyms are not provided. Is there any reason for this? Otherwise I would recommend the authors provide full forms of other acronyms too. It would make easier for a reader to follow.
- Line 276-281: “The cell wall 276 structure was relatively intact, with only a few stray microfibrils, and there were noswelling deformation of the chlorophylls and vacuole therein and no apparent plasmolysis. This indicated the cell wall degradation in the three Hami melon landraces (P < 279 0.05). Meanwhile, there were differences in cell wall degradation among the three landraces (P < 0.05).” Not clear to a reader. The authors should make it clearer.
- Line 283-284: “On day 0, the content of CWM of ‘CG’ was higher than that of ‘XZM 17’ and ‘JHM 25’ (P < 0.05)”. This statement is not true based on the data provided on Table 1 values with same subscript in a column are not statistically different. Further in Table 1, on CWM column why is standard deviation is zero. Is this really correct? Same for column “LC” with standard deviation zero.
- Figure 2: I would suggest the authors provide larger figures to make them more legible and remove the table which has currently been included in each figure as an inset. It cause a lot of distraction. The authors can move the information in table to the supplementary information.
- Figure 3 is not readable at all. Needs to be CHANGED.
Reviewer 3 Report
Zhang et al. have investigated the role of cell wall polysaccharides disassembly and enzyme activity changes in the softening process of Hami melon. The authors have made a detailed study to attain the proposed objective and the work is interesting to provide some significant correlations opening way for further study. However, the following points need to be addressed before this paper could be accepted for publication:
- L11-12 – “were studied during different storage stages of Hami melon…” should be corrected as “were studied with three Hami melon varieties representing three different storability …”.
- L16-19 – “…the degradation of pectin polysaccharides was obvious…the softening process of Hami melon.” should be corrected as “…the degradation of pectin polysaccharides was obvious involving depolymerization of macromolecular polymers accompanied by the production of new macromolecular polymers and composition change in pectin monosaccharides…the softening process of Hami melon.”
- L20-21 – “were related to fruit softening” should be corrected as “were associated with fruit softening”.
- L75 – “Fifty-five fruit” should be “Fifty-five fruits”.
- L96 – Is FW the fresh weight?
- L164-165 – please double-check the GC column dimension and include the film thickness.
- Section 3.2 numbering is missing.
- Figure 2 – inset table in A, B, C & E parts are not visible at all. If it contains significant data to shown, then please provide them separately in a table and delete them here. If not significant at all, please remove them to provide only the MW distribution curves.
- Figure 2 – The figure should be enlarged for better readability of legend labels. Also, provide the full form of “norm.”
- Figures 3 & 5 – these figures should be seriously enlarged to make the axes labels, legends, significance labels, error bars etc. for clarity and readability.
- L461-464 & L484-490 – these correlation data presentations here in the text looks clumsy and therefore it should be tabulated in a new Table 2.
- All the abbreviations should be carefully checked to provide full form in the first instance and abbreviated thereafter. Also, please avoid using abbreviations in sub-headings throughout the manuscript.
- Figures & Tables – All the abbreviations used should be provided in full form in the respective table footnotes and figure captions.
- As the authors made a detailed discussion of their study (which is of course good), various experimental components, work flow and trends/correlations observed in this study can be represented in a schematic figure for authors to have a quick understanding of the work in one glance.
